# Electromyographic measures of asymmetric muscle control of swallowing in Parkinson's disease

**Kasandra Diaz** *, **Elizabeth E. L. Stegemöller**

Department of Kinesiology, Iowa State University, Ames, IA, United States of America

* kdiaz@iastate.edu

## Abstract

### Introduction

During the early stages, Parkinson's disease (PD) is well recognized as an asymmetric disease with unilateral onset of resting tremor with varying degrees of rigidity, and bradykinesia. However, it remains unknown if other impairments, such as swallowing impairment (i.e., dysphagia), also present asymmetrically.

### Purpose

The primary aim of this study was to examine muscle activity associated with swallow on the most affected side (MAS) and least affected side (LAS) in persons with PD. A secondary aim was to explore the relationship between differences in muscle activity associated with swallow and subjective reports of swallowing impairment and disease severity.

### Methods

Function of muscles associated with swallowing was assessed using surface electromyography placed over the right and left submental and laryngeal regions during three swallows for a THIN and THICK condition. The Swallowing Quality of Life (SWAL-QOL) questionnaire and the Unified Parkinson's Disease Rating Scale (UPDRS) were collected as measures of subjective swallow impairment and disease severity, respectively.

### Results

Thirty-five participants diagnosed with idiopathic PD and on a stable antiparkinsonian medication regimen completed this study. Results revealed no significant mean difference in muscle activity during swallow between the more and less affected side. For the laryngeal muscle region, a significant difference in coefficient of variation between the MAS and LAS was revealed for peak amplitude for the THIN swallow condition. For the laryngeal muscle region, a significant association was revealed between muscle activity and disease severity but not subjective reports of swallowing impairment.

**Data Availability Statement:** Per the IRB-approved protocol, any researcher interested in the data can make a request to Dr. Stegemöller, the study PI. Dr. Stegemöller can share de-identified data that was used for the analyses published in the paper for future use. There is no one external to the study

that is also approved by IRB to hold the data and respond to external requests for data access. Our data is backed up regularly and is stored in three forms – DVD, internal server, and an external cloud-based storage – all are approved by our local IRB. Thus, long-term access to the data is possible in the case of any catastrophic event. Dr. Stegemöller will be responsible for long-term data storage and availability. If Dr. Stegemoller were to leave ISU, the long-term data storage and availability would become under the control of the Chair of the Kinesiology Department. This individual, whoever it may be at the time, will then be responsible for the long-term data storage and availability for the duration of their appointment, and this would continue for all future Chairs of the Department. Provided are multiple authors' contact information: Kasandra Diaz Email: Kdiaz@iastate.edu Elizabeth E.L Stegemöller esteg@iastate.edu Although the authors cannot make their study's data publicly available at the time of publication, all authors commit to make the data underlying the findings described in this study fully available without restriction to those who request the data, in compliance with the PLOS Data Availability policy. For data sets involving personally identifiable information or other sensitive data, data sharing is contingent on the data being handled appropriately by the data requester and in accordance with all applicable local requirements.

**Funding:** This study was supported by the Parkinson Study Group and the Parkinson's Disease Foundation's Advancing Parkinson's Treatment Innovations and in part by Iowa State University Extension and Outreach and joint Iowa agricultural extension districts. The grants provided financial support for the purchase of materials and participant reimbursement. The funders had no role in study design, data collection and analysis, decision to publish, or preparation of the manuscript. The authors received no specific funding for this work.

**Competing interests:** The authors have declared that no competing interests exist.

## Conclusion

Superficially it appears that swallowing impairment present symmetrical during the early stages of PD, however, our variability data indicates otherwise. These results will be used to inform future studies in specific types of swallowing impairment (i.e., oral dysphagia, pharyngeal dysphagia, and esophageal dysphagia), disease progression, and overall asymmetry.

## Introduction

Parkinson's disease (PD) is the second most common neurodegenerative disorder worldwide, affecting approximately 1% of the population over the age of 60 and 1% to 3% of those over the age of 80 [1, 2]. During the early stages, PD is well recognized as an asymmetric disease with unilateral onset of resting tremor with varying degrees of rigidity, and bradykinesia. The disease clinical asymmetry is associated with more severe contralateral nigrostriatal degeneration [3, 4]. However, it remains unknown if other impairments, such as swallowing impairment (i.e., dysphagia), also present asymmetrically. According to the literature, dysphagia prevalence is over 80% in patients with PD [5]. Here, muscles and nerves that allow food and liquid to efficiently move through the throat and into the esophagus could be damaged and dysfunctional due to cerebral atrophy, deterioration in nerve function, or a decline in mass of the muscle located in the throat [6]. Some of the main consequences of dysphagia include aspiration to pneumonia, malnutrition, and dehydration [7, 8], which are potentially fatal complications [9]. Within the central nervous system, the swallowing centers are bilaterally represented [10], and dysfunction within dopamine-related pathways have been implicated in the pathogenesis of dysphagia [11]. However, the dopamine precursor medication, which improves asymmetric motor symptoms of tremor, rigidity, and bradykinesia, appears to be an ineffective treatment for dysphagia [12, 13]. This suggests that the pathophysiology of PD dysphagia may differ, thus, there is a need to understand if dysphagia presents asymmetrically to inform future research on treatment and therapy strategies for persons with PD.

Videofluoroscopy swallowing study (VFSS) has been the traditional gold standard for evaluating dysphagia, as it allows for the direct assessment of the oral cavity, pharynx, and esophagus during swallowing and speech [8]. Studies using VFSS have shown a 75–100% incidence of dysphagia in individuals with PD [14]. In particular, VFSS revealed that patients with PD have difficulty initiating a swallow. This difficulty is marked by abnormal bolus formation during the oral phase and an inability to swallow, coughing, or aspiration during the pharyngeal phase [15]. Although VFSS provides images that may identify structural and motility abnormalities, to our knowledge, no studies have been conducted to study muscle activity and symptom asymmetry with VFSS. Thus, the use of non-invasive techniques to evaluate additional mechanisms involved during swallow may provide additional information.

Within the last decade, electromyography (EMG) has become a reliable and noninvasive [16] technique to evaluate the physiology of swallowing muscles amongst individuals with dysphagia. By using EMG, researchers were able to identify differences in swallowing muscle characteristics between PD and healthy individuals, such as prolonged muscle activity in persons with PD [17]. Furthermore, EMG can be used to characterize swallowing even when participants do not have significant problems with swallowing [18]. EMG has been also used to assess changes in swallow after a therapeutic intervention [19]. Taken together, EMG may be a valuable technique to assess swallow impairment and asymmetry in persons with PD.

Given that EMG is measured over separate muscle regions on each side of the neck, this technique may be used to determine if there is asymmetric activity during swallow in persons with PD. Thus, the purpose of this study was to examine muscle activity associated with swallow from the submental and laryngeal muscle regions on the more and less affected side in persons with PD. We hypothesized that since motor impairments in PD present asymmetrically, muscle activity associated with swallow would also be asymmetric in persons with PD. Additionally, we explored the relationship between muscle activity associated with swallow and subjective reports of swallowing impairment using the Swallowing Quality of Life (SWAL-QOL) questionnaire and disease severity using the Unified Parkinson's Disease Rating Scale (UPDRS). We hypothesized that the magnitude of difference in muscle activity between sides during swallowing might be more associated with 1) subjective report of swallow impairment (higher SWAL-QOL scores) and 2) disease severity (higher UPDRS scores).

## Patients and methods

### Participants

Thirty-five participants (57% female) diagnosed with idiopathic PD (mean age 67.7 ± 7.9 years) and on a stable antiparkinsonian medication regimen completed this study. The average duration of the disease was 7.8 years. Participants were excluded from the study if they presented significant cognitive impairment (Mini Mental State Exam score < 24), major psychiatric disorder (Beck Depression Inventory score < 18), untreated hypertension, or a history of head or neck cancer. No participant reported a diagnosis of dysphagia. Participants completed all study protocol on medication, approximately 1 to 1.5 hours after taking their normal dose of antiparkinsonian medication. All participants provided written informed consent, and the Iowa State University Institutional Review Board approved the study. The demographic and clinical data are shown in Table 1.

### More versus less affected side

To determine the most affected side, the Unified Parkinson's disease Rating Scale (UPDRS) was collected prior to beginning the swallow collections. A rater trained in the scoring the UPDRS completed the scoring at the time of collection. The most affected side (MAS) and least affected side (LAS) was determined by summing the scores for rigidity, bradykinesia, and

**Table 1. Participant demographics.**

| | |
|---|---|
| Gender (%F) | 57 |
| Age (yr.) | 67.7 ± 7.9 |
| Education (yr.) | 16.2 ± 3.5 |
| Disease Duration (yr.) | 7.8 ± 5.8 |
| Most Affected side (%R) | 37 |
| UPDRS | 57.6 ± 19.1 |
| Motor UPDRS | 29.3 ± 11.8 |
| MMSE | 28.9 ± 1.2 |
| BDI | 9.3 ± 5.5 |
| SWAL-QOL | 77.3±12.5 |

All values are presented as mean ± standard deviation. F = female; yr = years; UPDRS = Unified Parkinson's Disease Rating Scale; MMSE = Mini Mental State Exam; BDI = Beck Depression Inventory; SWAL-QOL = Swallowing Quality of Life Questionnaire. N = 35

tremor for the right and left side [20]. The side with the higher scores indicated more severe impairment, thus the side with the highest score was determined as the MAS. This was confirmed with participant self-report.

## EMG data collection and analysis

Swallowing function was assessed using surface EMG placed over the right and left submental and laryngeal muscle regions during three swallows for a THIN (10 mL of water) and THICK condition (10 mL of pudding) (see S1 Fig). For all six swallowing trials, either 10 mL of water or pudding was placed in front of the subject, and he or she were told to hold the bolus in their mouth and then swallow when they heard the auditory cue. The THIN and THICK conditions were not randomized. Participants first completed three swallow trials with the thin liquid and then three swallow trials with the thick pudding.

EMG (Delsys Trigno) output signals were recorded using the Motion Monitor software (Innovative Sports Training, Inc., Chicago IL) and sampled at 2000 Hz. A low pass filter at 500 Hz, a high pass filter at 1 Hz, and a notch filter at 60 Hz were applied. The raw signal was DC-corrected; full-wave rectified and smoothed using a root-mean-square envelope of 50 ms. The EMG signal was manually inspected for artifacts [19]. EMG measures from the submental and laryngeal muscle regions of the MAS and LAS included area under the curve, peak amplitude, onset time, offset time, rise time, and fall time. Area under the curve was calculated by approximating the region under the graph as a trapezoid and calculating its area. Peak amplitude was obtained as the peak in EMG activity. The duration of the EMG activity from time to onset to time of peak amplitude was used to calculate rise time. To calculate fall time, the duration of the EMG activity from time to peak amplitude to time to offset was used [19]. A quick inspection of the data revealed no differences in EMG outcome measure of onset, offset, rise, and fall time ($p > 0.05$). Therefore, coefficient of variation ($CV = SD/mean \times 100\%$) was only used to describe within-subject variability for area under the curve and peak amplitude for the MAS and LAS of the laryngeal and submental muscle regions during THICK and THIN swallow conditions. Mean scores from the three trials were calculated for both swallow conditions and muscle regions separately. Outliers that were two standard deviations away from the mean were excluded from the final statistical analysis. For each subject, a difference score was calculated by subtracting the LAS from the MAS for each EMG outcome measure. The methods used for evaluating swallowing function using EGM have been described previously [19].

## Swallow quality of life

The Swallowing Quality of Life (SWAL-QOL) questionnaire was collected to obtain a subjective evaluation of swallowing impairment. SWAL-QOL is a 44-item questionnaire using Likert-scale ratings to assess the impact of dysphagia on individuals' quality of life. SWAL-QOL scores ranged from 0 to 100, with a score of 100 representing no impairment [21]. Scores for each domain (i.e., food selection, burden, mental health, social functioning, fear, eating duration, eating desire, communication, sleep, and fatigue) were calculated and expressed as a percentage of the maximum possible points in the corresponding domain. For the total score, each domain score was summed and divided by 10 to produce an overall summary of QOL related to dysphagia [21].

## Statistical analysis

Mean ± standard error (SE) of all EMG outcome measures were calculated across all participants. Normality was assessed using the Shapiro-Wilk test. Due to the non-normal distribution of the EMG data, a Wilcoxon rank-sum test was applied to determine if there were differences

between the MAS and LAS for each EMG outcome measure (means and CV for area under the curve and peak EMG for the laryngeal and submental muscle regions during THICK and THIN swallow conditions) Significance was set at $\alpha < 0.05$. Due to non-normal distribution, a Spearman correlation was used to examine the associations between difference scores for each EMG outcome measure and 1) SWAL-QOL scores, and 2) the total UPDRS. To correct for multiple comparisons in our Spearman rank tests, the p-value threshold was defined using Bonferroni correction, resulting in a *p*-value threshold of $p < 0.025$. Statistical analysis was performed with IBM SPSS Statistics for Windows, Version 25.0.

## Results

Results revealed no significant difference between the MAS and LAS of the submental (see Table 2, Fig 1, and S1 Table) and laryngeal (see Table 2, Fig 2, and S1 Table) muscle regions in either swallow condition for EMG area under the curve, peak amplitude, rise time, and fall time ($p > 0.05$). No significant difference in coefficient of variation was revealed for the submental muscle region for the area under the curve and peak amplitude on the MAS and LAS for both swallow ($p > 0.05$) conditions (Fig 3). However, for the laryngeal muscle region, a significant difference between the MAS and LAS was revealed for peak amplitude for the THIN swallow ($p < 0.02$) condition (see Table 2 and Fig 4). No other significant differences in coefficient of variation for the laryngeal muscle region were revealed ($p > 0.07$).

No significant associations were revealed between difference scores in any of the EMG outcome measures and SWAL-QOL for either muscle region during the THICK or THIN swallow condition ($p > 0.04$). No significant associations were revealed for difference scores in any of the EMG outcome measures and UPDRS scores for either muscle region during the THICK swallow condition ($p > 0.18$). However, for the laryngeal muscle region, a significant association was revealed for difference scores for area under the curve and total UPDRS scores ($p = 0.003$) during the THIN swallow condition, but not for peak amplitude ($p = 0.315$). There was no significant association between UPDRS score and difference scores for either EMG

**Table 2. Statistical results for differences between the more and less affected side.**

| Most Affected Vs. Least Affected Side (mean) | Swallow Conditions | Z Score | P-Value |
|---|---|---|---|
| Laryngeal Area Under the Curve | THICK | -.020 | 0.84 |
| Laryngeal Area Under the Curve | THIN | -.742 | 0.46 |
| Submental Area Under the Curve | THICK | -.207 | 0.84 |
| Submental Area Under the Curve | THIN | -.957 | 0.34 |
| Laryngeal Peak Value | THICK | -.219 | 0.83 |
| Laryngeal Peak Value | THIN | -.447 | 0.66 |
| Submental Peak Value | THICK | -1.892 | 0.06 |
| Submental Peak Value | THIN | -.765 | 0.44 |
| **Most Affected Vs. Least Affected Side (CV)** | **Swallow Conditions** | **Z Score** | **P-Value** |
| Laryngeal Area Under the Curve | THICK | -0.698 | 0.49 |
| Laryngeal Area Under the Curve | THIN | -1.825 | 0.07 |
| Submental Area Under the Curve | THICK | -0.639 | 0.52 |
| Submental Area Under the Curve | THIN | -1.095 | 0.27 |
| Laryngeal Peak Value | THICK | -0.760 | 0.45 |
| Laryngeal Peak Value | THIN | -2.372 | 0.02 |
| Submental Peak Value | THICK | -1.217 | 0.22 |
| Submental Peak Value | THIN | -0.061 | 0.95 |

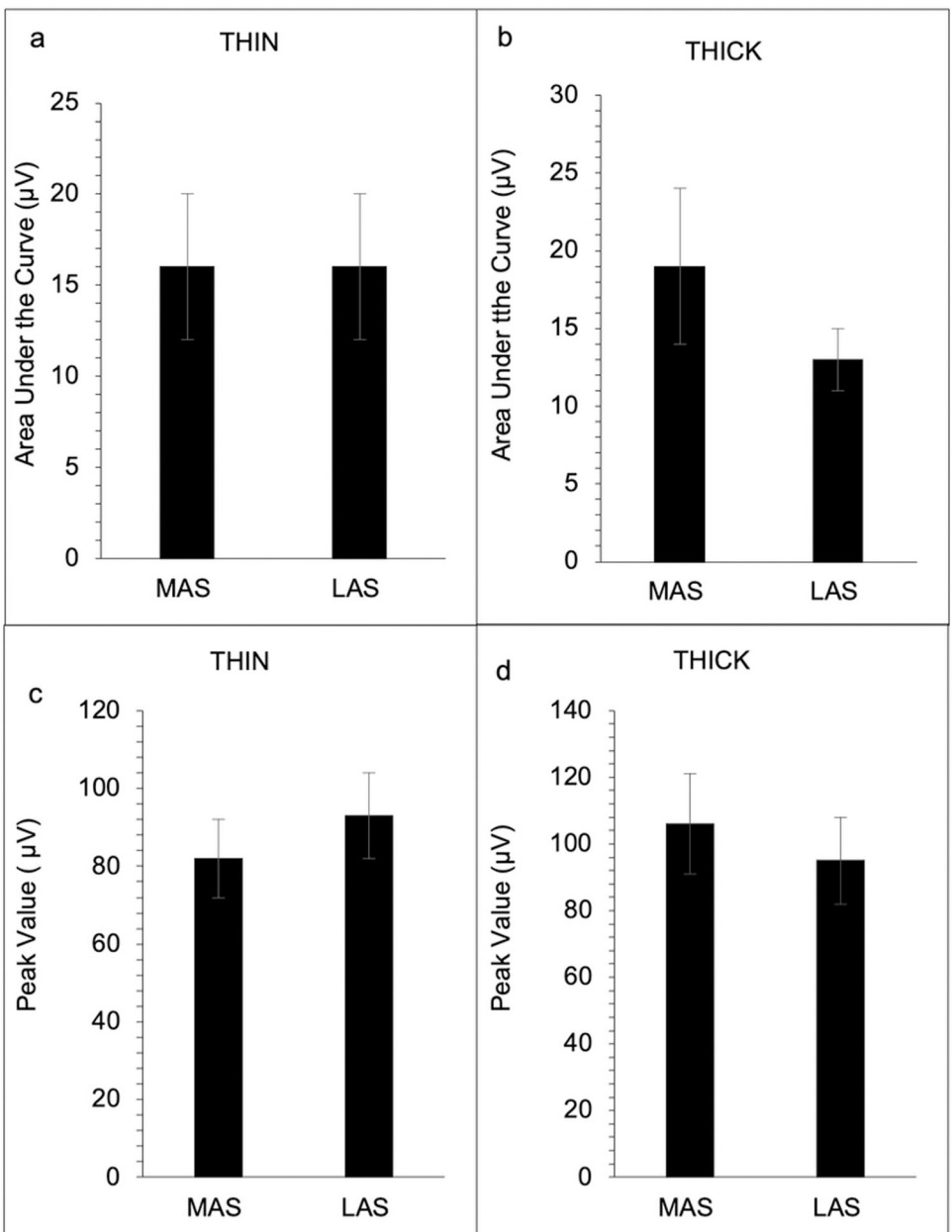

**Fig 1. EMG measures for submental muscles groups for Most Affected Side (MAS) and Least Affected Side (LAS) across both swallow conditions (N = 23).** Area under the curve for the submental muscle group for the thin swallow condition (a), and area under the curve for the submental group for the thick swallow condition (b). Peak amplitude for the submental muscle group during the thin swallow condition (c), and peak amplitude for the submental group during the thick swallow condition (d). Error bars reflect standard error.

outcome measure for the submental muscle region during the THIN swallow ($p > 0.852$) condition (Table 3).

## Discussion

The present study assessed EMG measures from the submental and laryngeal muscle regions of the more and less affected side to determine if swallowing impairment presents

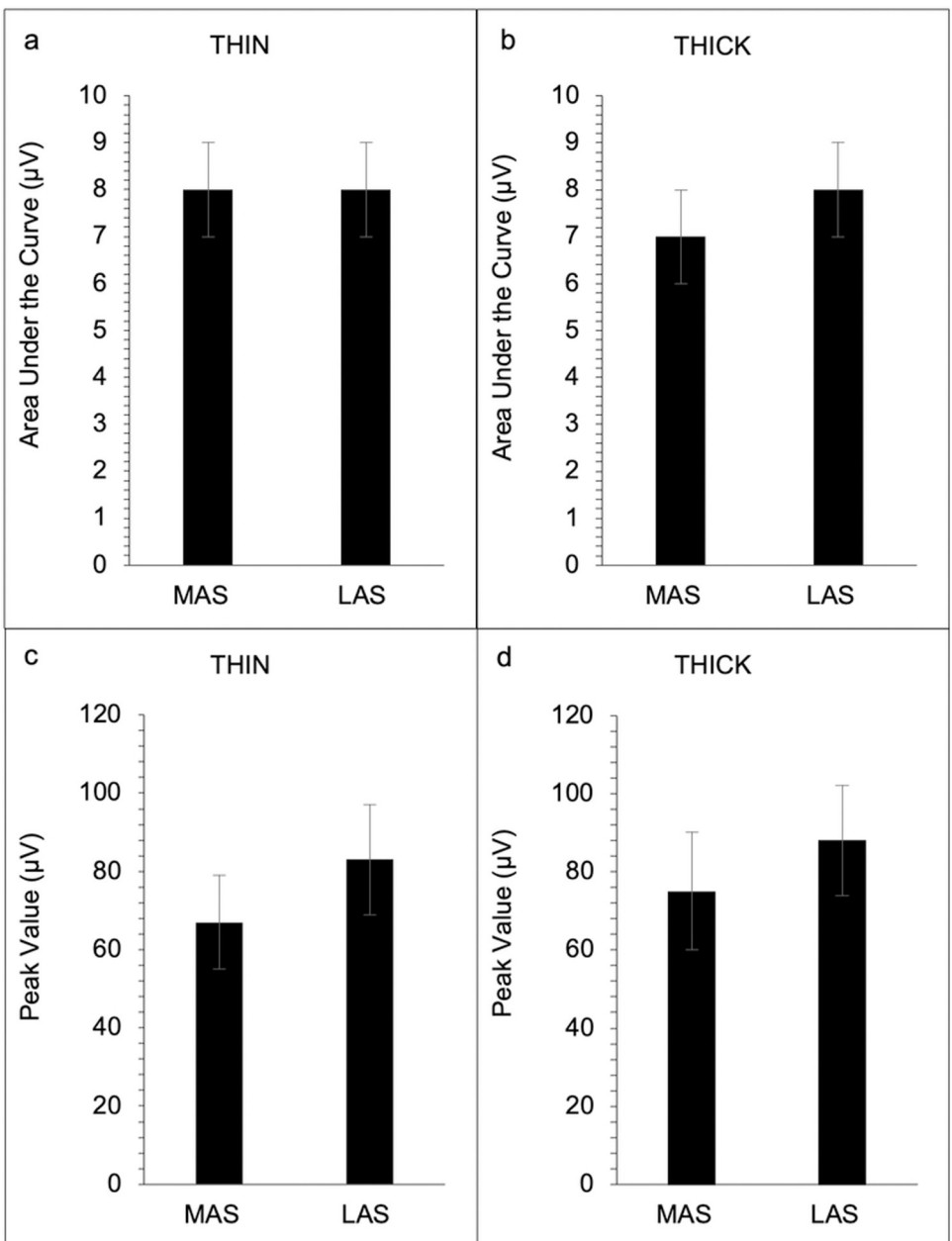

**Fig 2. EMG measures for laryngeal muscles groups for Most Affected Side (MAS) and Least Affected Side (LAS) across both swallow conditions (N = 23).** Area under the curve for the laryngeal muscle group for the thin swallow condition (a), and area under the curve for the laryngeal group for the thick swallow condition (b). Peak amplitude for the laryngeal muscle group during the thin swallow condition (c), and peak amplitude for the laryngeal muscle group during the thick swallow condition (d). Error bars reflect standard error.

asymmetrically in persons with PD. Associations between the magnitude of muscle activity and the SWAL-QOL and UPDRS were are also assessed to determine if there was a relationship between 1) subject reports of swallow impairment and EMG activity and 2) disease severity and EMG activity. We hypothesized that because PD is an asymmetric disease than EMG activity associated with swallow would also be asymmetric. We also hypothesized that the magnitude of difference between sides might be more associated with higher UPDRS and

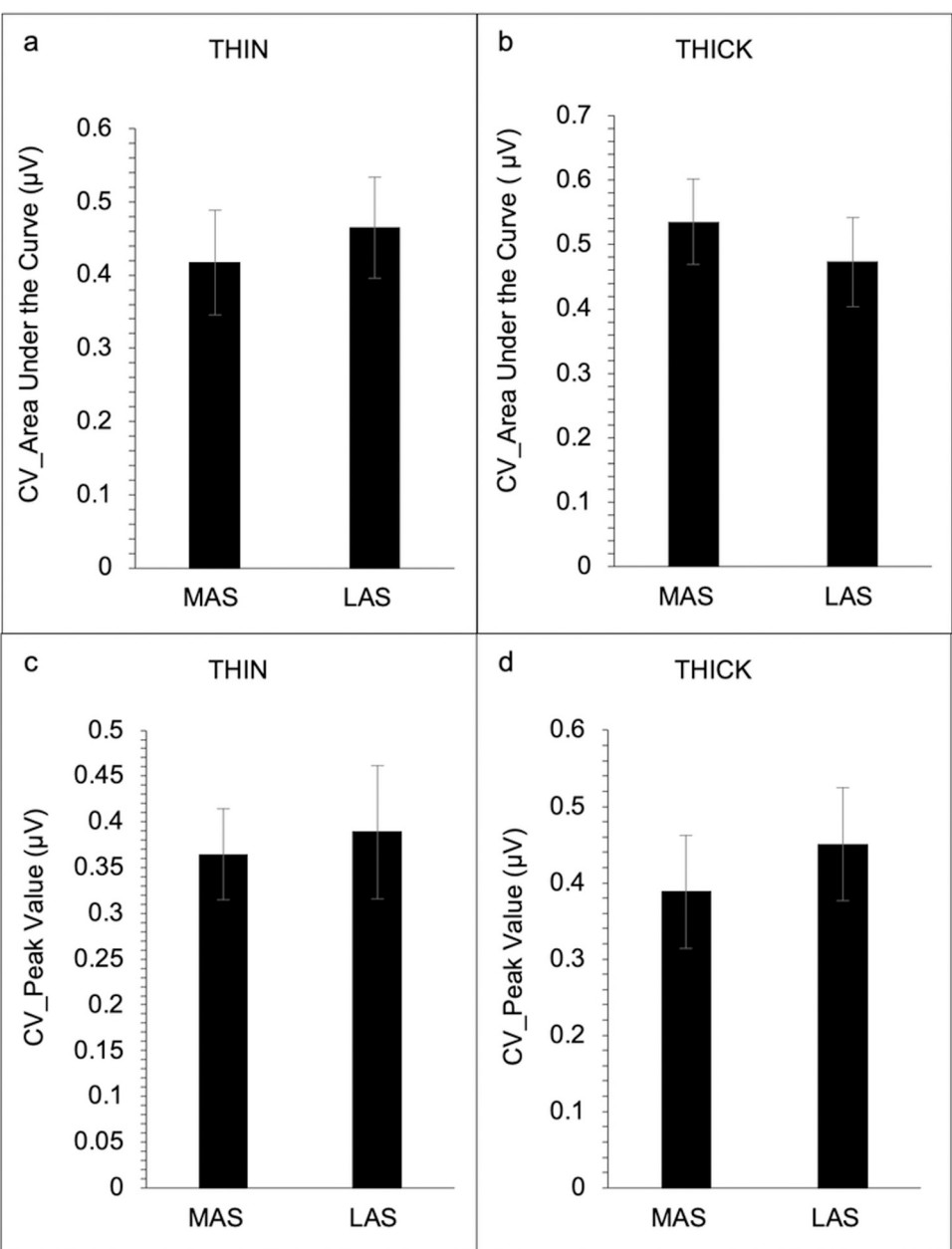

**Fig 3. Coefficient of variation on EMG measures for submental muscles groups for Most Affected Side (MAS) and Least Affected Side (LAS) across both swallow conditions (N = 23).** Area under the curve for the submental muscle group for the thin swallow condition (a), and area under the curve for the submental group for the thick swallow condition (b). Peak amplitude for the submental muscle group during the thin swallow condition (c), and peak amplitude for the submental group during the thick swallow condition (d). Error bars reflect standard error.

SWOL-QOL scores. Our main findings did not support our swallow asymmetry hypothesis. Our results revealed no significant difference between the MAS and LAS of the submental and laryngeal muscle regions in either swallow condition for any of the EMG outcome measures. However, for variability, a significant difference between the MAS and LAS was revealed for the peak amplitude during the THIN swallow condition for the laryngeal muscle region. For our hypothesis regarding swallowing asymmetry and higher scores on the SWAL-QOL and

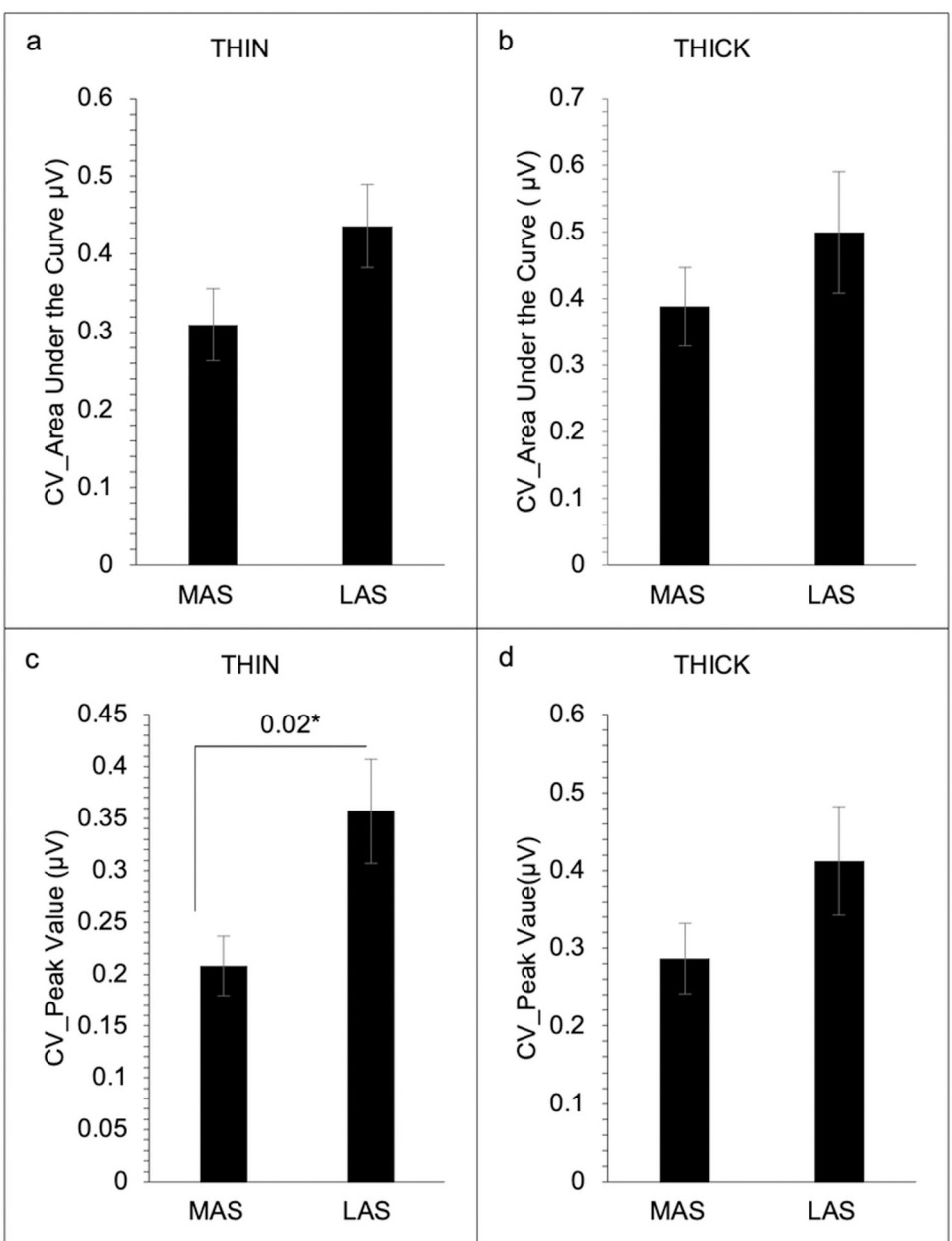

**Fig 4. Coefficient of variation on EMG measures for laryngeal muscles groups for Most Affected Side (MAS) and Least Affected Side (LAS) across both swallow conditions (N = 23).** Area under the curve for the laryngeal muscle group for the thin swallow condition (a), and area under the curve for the laryngeal group for the thick swallow condition (b). Peak amplitude for the laryngeal muscle group during the thin swallow condition (c), and peak amplitude for the laryngeal muscle group during the thick swallow condition (d). Error bars reflect standard error.

UPDRS, our findings partially support this hypothesis, revealing a significant association between the difference scores for area under the curve and total UPDRS scores during the THIN swallow condition.

We hypothesized that since PD is an asymmetric disease, EMG activity associated with swallow would also be asymmetric in persons with PD. In contrast to our hypothesis, no significant difference was observed between the MAS and LAS of the submental and laryngeal

**Table 3. Statistical results for associations between EMG outcome measures, UPDRS scores, and SWAL-QOL scores.**

| Most Affected Vs. Least Affected Side | Swallow Conditions | SWAL-QOL | P-Values | UPDRS | P-Values |
|---|---|---|---|---|---|
| Submental Area Under the Curve | THICK | 0.014 | 0.948 | 0.017 | 0.939 |
| Submental Area Under the Curve | THIN | -0.266 | 0.220 | -0.001 | 0.996 |
| Submental Peak Amplitude | THICK | -0.4 | 0.059 | 0.286 | 0.186 |
| Submental Peak Amplitude | THIN | -0.339 | 0.113 | -0.041 | 0.852 |
| Laryngeal Area Under the Curve | THICK | -0.043 | 0.845 | 0.123 | 0.577 |
| Laryngeal Area Under the Curve | THIN | 0.422 | 0.045 | -0.587 | 0.003 |
| Laryngeal Peak Amplitude | THICK | -0.117 | 0.596 | -0.034 | 0.877 |
| Laryngeal Peak Amplitude | THIN | 0.186 | 0.397 | -0.219 | 0.315 |

*p* Values were calculated using Spearman correlation coefficient (rho). After correcting for multiple comparisons, significance was set at p > 0.025. Light grey shades indicate trends and dark grey shades indicate significant associations.

muscle regions in either swallow condition for EMG outcome measures. A possible explanation for this finding is that unlike other movement, regardless of level of impairment, bilateral musculature is recruited during swallowing to safely guide food bolus past the airway and into the esophagus. Since severe dysphagia is commonly present in advanced patients with PD, it may be possible that in prodromal stages of the disease, swallowing asymmetry is present. However, by the time diagnosis occurs, and appendicular asymmetries are present, both sides of cranial structures may have advanced sufficiently that asymmetries associated with swallow muscles are no longer detectable. Finally it is also possible that this finding is attributed to some technical constraints as EMG electrodes are not able to reliably distinguish swallow from submental muscle activation that occurs when stabilizing the floor of mouth during tongue movements [22]. That is to say, that the EMG activity reported in our study may be related to tongue movement, resulting in a signal that is not truly representative of the MAS and LAS of the submental and laryngeal muscle regions. Thus, to visualize and examine swallowing symmetry in persons with PD, future studies should use a combination of videofluoroscopic examination and EMG techniques.

For variability, a significant difference between the MAS and LAS was revealed for the peak amplitude during the THIN swallow condition for the laryngeal muscle region. Specifically, the data revealed reduced variability in the MAS when compared to the LAS. Although other studies have associated impairment with increased movement variability [23, 24], in our case, decrease muscle variability may indicate a reduction in muscle force production in the MAS, as previous research has shown that muscle activity is reduced and disrupted in patients with PD [25–27]. Although our data show a trend towards a decrease in muscle amplitude and variability in the MAS (Fig 4), the findings were not statistically significant across muscle regions and swallow conditions. This may be attributed to our small sample size and sample variability. Thus, follow-up studies with larger sample sizes are needed.

Next, we were interested in understanding if these differences demonstrate a relationship with subjective reports of swallow and disease severity. Differences between the MAS and LAS muscle activity during the THIN condition only was associated with disease severity. However, differences between the MAS and LAS muscle activity did not predict subjective reports of swallowing impairment. The results may be due to our sample having minimal to no swallowing impairment since they are early in disease severity (Table 1). Thus, future studies need to explore differences between dysphasic and non-dysphasic patients with PD.

## Limitations

Placement of the submental and laryngeal EMG requires considerable skill and precision, and it is possible that electrodes placements from person to person may have confounded results. However, every effort was made to place the EMG electrodes accurately. Furthermore, to control for this potential source of error we conducted within subject comparisons. Although EMG can reveal muscle dysfunction, for a more accurate representation of swallowing impairment in persons with PD, in future studies EMG should be coupled with other metrics, such as air flow, and accelerometry. The analysis was also limited to the subjective reports of swallowing impairment and did not consider the effect of any specific type of swallowing impairments (i.e., oral dysphagia, pharyngeal dysphagia, and esophageal dysphagia). However, this was beyond the scope of this study and will be considered in future experiments. Cued swallow is less likely to be representative of a typical swallow. Furthermore, because PD is not just a sensorimotor disorder, but also cognitive, and subject to higher-level cognitive input, it is possible that the cuing of swallows had a positive impact on swallowing performance, and thus overall symmetry. Amongst those that show asymmetry (22 participants), The range score for differences between sides was 1–9. It is possible that the variability in asymmetry may have washed out our results as the difference between the most and least affected side was less than 3 points in the MDS-UPDRS for 15 participants. Finally, per the SWAL-QOL scores, patients in this sample had very low levels of swallowing impairment, thus results may not be generalized to patients with severe swallowing impairment. The testing was also performed ON meds, which might have masked any latent asymmetries that would give patients problems during low periods of their medication cycles. Regardless, the results are intriguing and clearly warrant replication.

## Conclusions

Superficially it appears that swallowing impairment present symmetrical during the early stages of PD, however, our variability data indicates otherwise. This is important to consider clinically when predicting the natural course of swallowing impairment in persons with PD and individual treatment strategies. Future studies will involve a similar paradigm in combination with other metrics, such as air flow, and accelerometry to explore specific types of swallowing impairment (i.e., oral dysphagia, pharyngeal dysphagia, and esophageal dysphagia), disease progression, and overall asymmetry.

## Supporting information

**S1 Fig. Data collection set up.**
(DOCX)

**S1 Table. Statistical results for differences between the more and less affected side.**
(DOCX)

## Author Contributions

**Conceptualization:** Elizabeth E. L. Stegemöller.

**Data curation:** Kasandra Diaz, Elizabeth E. L. Stegemöller.

**Formal analysis:** Kasandra Diaz, Elizabeth E. L. Stegemöller.

**Funding acquisition:** Elizabeth E. L. Stegemöller.

**Investigation:** Kasandra Diaz, Elizabeth E. L. Stegemöller.

**Methodology:** Kasandra Diaz, Elizabeth E. L. Stegemöller.

**Project administration:** Elizabeth E. L. Stegemöller.

**Resources:** Elizabeth E. L. Stegemöller.

**Supervision:** Elizabeth E. L. Stegemöller.

**Writing – original draft:** Kasandra Diaz.

**Writing – review & editing:** Elizabeth E. L. Stegemöller.

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
