## [Decision Letter · Decision Letter 0]

8 Jul 2021

PONE-D-21-07951

Swallowing impairment in persons with Parkinson’s disease

PLOS ONE

Dear Dr. Diaz,

Thank you for submitting your manuscript to PLOS ONE. After careful consideration, we feel that it has merit but does not fully meet PLOS ONE’s publication criteria as it currently stands. Therefore, we invite you to submit a revised version of the manuscript that addresses the points raised during the review process.

We look forward to receiving your revised manuscript.

Kind regards,

J. Lucas McKay, Ph.D., M.S.C.R.

Academic Editor

PLOS ONE

“This study was supported by the Parkinson Study Group and the Parkinson’s Disease Foundation’s Advancing Parkinson’s Treatment Innovations and in part by Iowa State University Extension and Outreach and joint Iowa agricultural extension districts. “

We note that you have provided funding information that is not currently declared in your Funding Statement. However, funding information should not appear in the Funding section or other areas of your manuscript. We will only publish funding information present in the Funding Statement section of the online submission form.

“This study was supported by the Parkinson Study Group and the Parkinson’s Disease Foundation’s Advancing Parkinson’s Treatment Innovations and in part by Iowa State University Extension and Outreach and joint Iowa agricultural extension districts.”

“This study was supported by the Parkinson Study Group and the Parkinson’s Disease Foundation’s Advancing Parkinson’s Treatment Innovations and in part by Iowa State University Extension and Outreach and joint Iowa agricultural extension districts."

d) If you did not receive any funding for this study, please state: “The authors received no specific funding for this work.

Reviewers' comments:

Reviewer's Responses to Questions

**Comments to the Author**

1. Is the manuscript technically sound, and do the data support the conclusions?

Reviewer #1: Yes

Reviewer #2: Yes

2. Has the statistical analysis been performed appropriately and rigorously? 

Reviewer #1: Yes

Reviewer #2: Yes

3. Have the authors made all data underlying the findings in their manuscript fully available?

Reviewer #1: Yes

Reviewer #2: Yes

4. Is the manuscript presented in an intelligible fashion and written in standard English?

Reviewer #1: Yes

Reviewer #2: Yes

5. Review Comments to the Author

Reviewer #1: The authors examine whether muscle activity during swallowing is asymmetric in people with moderate Parkinson's disease. For the most part they do not find evidence of asymmetry, but they do identify a consistent pattern of the "more affected side" of PD patients exhibiting less variability across swallowing tasks.

This work is essentially a negative result but is nonetheless a worthwhile contribution to the literature. The stratified statistical analysis used was not the most efficient use of the data, as a linear model would have been more likely to demonstrate statistical significance, but the finding that the more affected side was LESS variable from trial to trial is interesting and novel. The plots in Figure 4 support the coherence of this finding across conditions, even though it only reached statistical significance in one stratum.

As presented, the study population is not well-suited to investigate this question. None of the patients had dysphagia, and they almost certainly had bilateral symptoms with ON-MEDS UPDRS motor scores of 29. If there had been a healthy control group, the authors might have been more likely to identify asymmetry in PD beyond that expected in age-matched individuals, but this was not the case. The testing was also performed ON meds, which might have masked any latent asymmetries that would give patients problems during low periods of their medication cycles.

The authors have done a fairly thorough job of addressing the various limitations of the work, and the technical presentation is generally adequate to enable replication, notwithstanding a few details below. The reviewer's main complaint is that the title "Swallowing impairment in persons with Parkinson’s disease" is not really representative of what was done. Something more appropriate might be "Electromyographic measures of asymmetric muscle control of swallowing in Parkinson's disease" or similar.

Similarly, there are several places throughout the manuscript where sentences like "there is a need to understand if dysphagia presents asymmetrically to inform treatment and therapy strategies for persons with PD" appear. That this need is there is true, but that is not the question that is being asked with this study design. Consider revising these sections.

A few technical notes follow.

1. What were the numerical values/ranges of asymmetry? It seems a score of 12/22 would be much more assymetric than, say, 10/12. If the patients were not particularly asymmetric then it might explain the absence of identified asymmetry. Stebbins and Goetz have suggested formulae for calculation of TD/PIGD phenotype from UPDRS scores, in which there is a substantial range of "Indeterminate" scores, a similar phenomenon might be at work here.

2. How were the EMG signals aligned to the cue to the subject? What variability was associated with this?

3. It is not completely clear how CV was calculated. Within each patient, AUC CV of the less affected side was subtracted from that of the more affected side, providing a single number for each patient?

4. The material in introduction on VFSS is pretty extensive, given that VFSS was not actually done here. Recommend reducing.

Reviewer #2: The study presented the results of primary scientific research:

this study aimed to examine muscle activity associated with swallow on the more and less affected side in persons with PD. A secondary aim was to explore the relationship between differences in muscle activity associated with

swallow and subjective reports of swallowing impairment and disease severity.

Experiments, statistics, and other analyses are performed to a high technical standard and are described in sufficient detail. Overall, more detail regarding methodology of administration of swallow trials is required to ensure transparency, replicability and ensure a correct interpretation of the results (e.g. was the administration of the THIN and THICK liquids randomized?).

The article is presented in an intelligible fashion and is written in standard English. Overall, the writing is very clear and succinct.

Minor comments:

1) Consider breaking some long sentences into two separate sentences. Some sentences are about 50 words in length.

2) missing words (i.e. "Swallowing Quality of (SWAL-QOL)" pp. 2)

3) provide the explanation of acronyms the first time they are written in the text (e.g. "MAS" and "LAS" are present in the abstact, but their explanation is provided at pp. 6)

6. PLOS authors have the option to publish the peer review history of their article (what does this mean?). If published, this will include your full peer review and any attached files.

Reviewer #1: No

Reviewer #2: No

---

## [Author Response · Author response to Decision Letter 0]

21 Oct 2021

RESPONSE: Thank you for pointing this out. The authors have made the necessary changes throughout the manuscript to ensure it meets PLOS ONE’S style requirements. 

“This study was supported by the Parkinson Study Group and the Parkinson’s Disease Foundation’s Advancing Parkinson’s Treatment Innovations and in part by Iowa State University Extension and Outreach and joint Iowa agricultural extension districts. “

We note that you have provided funding information that is not currently declared in your Funding Statement. However, funding information should not appear in the Funding section or other areas of your manuscript. We will only publish funding information present in the Funding Statement section of the online submission form.

“This study was supported by the Parkinson Study Group and the Parkinson’s Disease Foundation’s Advancing Parkinson’s Treatment Innovations and in part by Iowa State University Extension and Outreach and joint Iowa agricultural extension districts.”

RESPONSE: Thank you for pointing this out. The authors have made the necessary changes and have added this statement to the cover letter submitted with the manuscript. 

“This study was supported by the Parkinson Study Group and the Parkinson’s Disease Foundation’s Advancing Parkinson’s Treatment Innovations and in part by Iowa State University Extension and Outreach and joint Iowa agricultural extension districts."

RESPONSE: The grants provided financial support for the purchase of materials and participant reimbursement

RESPONSE: The funders had no role in study design, data collection and analysis, decision to publish, or preparation of the manuscript. 

c) If any authors received a salary from any of your funders, please state which authors and which funders. If you did not receive any funding for this study, please state: “The authors received no specific funding for this work.

RESPONSE: The authors received no specific funding for this work.

a) If there are ethical or legal restrictions on sharing a de-identified data set, please explain them in detail (e.g., data contain potentially sensitive information, data are owned by a third-party organization, etc.) and who has imposed them (e.g., an ethics committee). Please also provide contact information for a data access committee, ethics committee, or other institutional body to which data requests may be sent.b) If there are no restrictions, please upload the minimal anonymized data set necessary to replicate your study findings as either Supporting Information files or to a stable, public repository and provide us with the relevant URLs, DOIs, or accession numbers. For a list of acceptable repositories, please see http://journals.plos.org/plosone/s/data-availability#loc-recommended-repositories.We will update your Data Availability statement on your behalf to reflect the information you provide.

RESPONSE: We apologize, but at the time of IRB approval for this study (2013), we did not obtain approval to publicly share de-identified data. The study is now closed as all data is complete, de-identified, and in the publication stage. There is not way to request a modification to share the date, so we used the statement that the data is available upon request. 

Reviewer #1: 

The authors examine whether muscle activity during swallowing is asymmetric in people with moderate Parkinson's disease. For the most part they do not find evidence of asymmetry, but they do identify a consistent pattern of the "more affected side" of PD patients exhibiting less variability across swallowing tasks.

This work is essentially a negative result but is nonetheless a worthwhile contribution to the literature. The stratified statistical analysis used was not the most efficient use of the data, as a linear model would have been more likely to demonstrate statistical significance, but the finding that the more affected side was LESS variable from trial to trial is interesting and novel. The plots in Figure 4 support the coherence of this finding across conditions, even though it only reached statistical significance in one stratum.

As presented, the study population is not well-suited to investigate this question. None of the patients had dysphagia, and they almost certainly had bilateral symptoms with ON-MEDS UPDRS motor scores of 29. If there had been a healthy control group, the authors might have been more likely to identify asymmetry in PD beyond that expected in age-matched individuals, but this was not the case. The testing was also performed ON meds, which might have masked any latent asymmetries that would give patients problems during low periods of their medication cycles.

RESPONSE: We thank the reviewer for the insightful review of this manuscript. We agree with the points made above and have included them in the limitations section. (Page. 16, Lines 339-346)

The authors have done a fairly thorough job of addressing the various limitations of the work, and the technical presentation is generally adequate to enable replication, notwithstanding a few details below. The reviewer's main complaint is that the title "Swallowing impairment in persons with Parkinson’s disease" is not really representative of what was done. Something more appropriate might be "Electromyographic measures of asymmetric muscle control of swallowing in Parkinson's disease" or similar.

RESPONSE: This is a good suggestion. We have taken the reviewer comments into consideration and made the appropriate changes to the title of the manuscript. 

Similarly, there are several places throughout the manuscript where sentences like "there is a need to understand if dysphagia presents asymmetrically to inform treatment and therapy strategies for persons with PD" appear. That this need is there is true, but that is not the question that is being asked with this study design. Consider revising these sections.

RESPONSE: The authors of the manuscript acknowledge the reviewer comment and have revised those sections to note that our research study is important as it could inform future research on treatment and therapy strategies for persons with PD. (Page 3, Line 85)

A few technical notes follow.

1. What were the numerical values/ranges of asymmetry? It seems a score of 12/22 would be much more asymmetric than, say, 10/12. If the patients were not particularly asymmetric then it might explain the absence of identified asymmetry. Stebbins and Goetz have suggested formulae for calculation of TD/PIGD phenotype from UPDRS scores, in which there is a substantial range of "Indeterminate" scores, a similar phenomenon might be at work here.

RESPONSE: This is a great point. The range score for differences between sides using the MDS-UPDRS was 0-9. 13 out of 35 participants show no-symmetry, and of those that did show asymmetry (22 participants), the range score for the MDS-UPDRS was 1-9. Out of these 22, the difference between the most and least affected side was less than 3 points in 15 of the participants. It is possible that this variability in asymmetry may have washed out our results and is included in the limitation section (Page16, Lines 339-342). We did complete the calculations following Stebbins et al., (2013) as suggested. The formulae for calculation of TD/PIGD phenotype from UPDRS and MDS-UPDRS scores revealed that 22 out of 35 participants fell under the PIGD phenotype, 12 were TD, and only 1 participant fell under the indeterminate category. Thus, we feel that it is unlikely that the indeterminate is affecting our results; therefore, we did not include it in the manuscript as this time. If the reviewer feels this is needed in the manuscript, we are happy to consider further revision. 

2. How were the EMG signals aligned to the cue to the subject? What variability was associated with this? 

RESPONSE: Thank you for these insightful questions. Participants were directed to hold the bolus in their mouth and to swallow upon hearing an auditory cue. This is now included in the methods (Page 7, Line 158- 161). As part of our data analysis, we calculated onset time, the variability associated with onset time was not significant amongst participants (p >0.05) this information is now included in the methods (page 7, Line 175-176). 

3. It is not completely clear how CV was calculated. Within each patient, AUC CV of the less affected side was subtracted from that of the more affected side, providing a single number for each patient?

RESPONSE: This is correct. Within each patient, a difference score was calculated by subtracting the LAS from the MAS for each EMG outcome measure. This has been clarified in the methods. (Page 7, Line 175-193)

4. The material in introduction on VFSS is pretty extensive, given that VFSS was not actually done here. Recommend reducing.

RESPONSE: We have tried to reduce this section; however, we have received information from previous reviewers wanting this information in the manuscript. (Page 4, Line 91-100)

Reviewer #2:

The study presented the results of primary scientific research:

this study aimed to examine muscle activity associated with swallow on the more and less affected side in persons with PD. A secondary aim was to explore the relationship between differences in muscle activity associated with swallow and subjective reports of swallowing impairment and disease severity.

Experiments, statistics, and other analyses are performed to a high technical standard and are described in sufficient detail. Overall, more detail regarding methodology of administration of swallow trials is required to ensure transparency, replicability and ensure a correct interpretation of the results (e.g. was the administration of the THIN and THICK liquids randomized?).

RESPONSE: Thank you for your positive review! Participants were directed to hold the bolus in their mouth and to swallow upon hearing an auditory cue. The administration of the THIN and THICK liquids was not randomized. Participants first swallow the thin liquid and then the thick pudding. This is included in the methods (Page 7, Lines 157-161). 

The article is presented in an intelligible fashion and is written in standard English. Overall, the writing is very clear and succinct.

REPONSE: Thank you.

Minor comments:

1) Consider breaking some long sentences into two separate sentences. Some sentences are about 50 words in length.

RESPONSE: Thank you for pointing this out. We have made the necessary changes throughout the manuscript. 

2) Missing words (i.e., "Swallowing Quality of (SWAL-QOL)" pp. 2)

RESPONSE: Thank you again. We have added the missing words (Page 2, Line 52). 

3) Provide the explanation of acronyms the first time they are written in the text (e.g. "MAS" and "LAS" are present in the abstract, but their explanation is provided at pp. 6)

RESPONSE: We apologize for this oversight. We have added an explanation of the acronyms in the written text (Page 2, Line 47).

---

## [Decision Letter · Decision Letter 1]

26 Dec 2021

Electromyographic measures of asymmetric muscle control of swallowing in Parkinson's disease

PONE-D-21-07951R1

Dear Dr. Diaz,

We’re pleased to inform you that your manuscript has been judged scientifically suitable for publication and will be formally accepted for publication once it meets all outstanding technical requirements.

Kind regards,

Jianhong Zhou

Staff Editor

PLOS ONE

Additional Editor Comments (optional):

Reviewers' comments:

Reviewer's Responses to Questions

**Comments to the Author**

1. If the authors have adequately addressed your comments raised in a previous round of review and you feel that this manuscript is now acceptable for publication, you may indicate that here to bypass the “Comments to the Author” section, enter your conflict of interest statement in the “Confidential to Editor” section, and submit your "Accept" recommendation.

Reviewer #1: All comments have been addressed

Reviewer #2: All comments have been addressed

2. Is the manuscript technically sound, and do the data support the conclusions?

Reviewer #1: Yes

Reviewer #2: Yes

3. Has the statistical analysis been performed appropriately and rigorously? 

Reviewer #1: Yes

Reviewer #2: (No Response)

4. Have the authors made all data underlying the findings in their manuscript fully available?

Reviewer #1: Yes

Reviewer #2: (No Response)

5. Is the manuscript presented in an intelligible fashion and written in standard English?

Reviewer #1: Yes

Reviewer #2: Yes

6. Review Comments to the Author

Reviewer #1: I have no further comments. The authors have addressed the concerns. I have no further comments. The authors have addressed the concerns.

Reviewer #2: The authors have adequately addressed the comments raised in the previous round of review.

Minor comment: Beware of typing errors such as uppercase and lowercase, es. "The Swallowing Quality of life (SWAL-QOL)" (Page 2, Line 50).

7. PLOS authors have the option to publish the peer review history of their article (what does this mean?). If published, this will include your full peer review and any attached files.

Reviewer #1: No

Reviewer #2: No

---

## [Editor Report · Acceptance letter]

28 Jan 2022

PONE-D-21-07951R1 

Electromyographic measures of asymmetric muscle control of swallowing in Parkinson's disease 

Dear Dr. Diaz:

I'm pleased to inform you that your manuscript has been deemed suitable for publication in PLOS ONE. Congratulations! Your manuscript is now with our production department. 

Kind regards, 

on behalf of

Dr. J. Lucas McKay 

Academic Editor

PLOS ONE